# Genes and Dietary Fatty Acids in Regulation of Fatty Acid Composition of Plasma and Erythrocyte Membranes

**DOI:** 10.3390/nu10111785

**Published:** 2018-11-16

**Authors:** Maria Lankinen, Matti Uusitupa, Ursula Schwab

**Affiliations:** 1Institute of Public Health and Clinical Nutrition, University of Eastern Finland, 70211 Kuopio, Finland; matti.uusitupa@uef.fi (M.U.); ursula.schwab@uef.fi (U.S.); 2Department of Medicine, Endocrinology and Clinical Nutrition, Kuopio University Hospital, 70210 Kuopio, Finland

**Keywords:** fatty acid, diet, genotype, human, *FADS*

## Abstract

The fatty acid compositions of plasma lipids and cell membranes of certain tissues are modified by dietary fatty acid composition. Furthermore, many other factors (age, sex, ethnicity, health status, genes, and gene × diet interactions) affect the fatty acid composition of cell membranes or plasma lipid compartments. Therefore, it is of great importance to understand the complexity of mechanisms that may modify fatty acid compositions of plasma or tissues. We carried out an extensive literature survey of gene × diet interaction in the regulation of fatty acid compositions. Most of the related studies have been observational studies, but there are also a few intervention trials that tend to confirm that true interactions exist. Most of the studies deal with the desaturase enzyme cluster (*FADS1*, *FADS2*) in chromosome 11 and elongase enzymes. We expect that new genetic variants are being found that are linked with the genetic regulation of plasma or tissue fatty acid composition. This information is of great help to understanding the contribution of dietary fatty acids and their endogenic metabolism to the development of some chronic diseases.

## 1. Introduction

Traditionally, the fatty acid compositions of plasma and its components (triglycerides (TG), phospholipids (PL), cholesteryl esters (CE)) have been used as biomarkers of dietary intake of certain fatty acids. Furthermore, the fatty acid compositions of adipose tissue and erythrocyte and platelet membranes reflect the dietary intakes of different fatty acids [1,2,3,4]. In particular, proportions of omega-3 long-chain unsaturated fatty acids in CE and in erythrocyte and platelet membranes reflect quite well the intakes of these fatty acids, e.g., from fatty fish. On the contrary, the major dietary saturated fatty acid, palmitic acid, is more problematic as a biomarker of saturated fatty acid intake due to its rapid desaturation and elongation to longer-chain fatty acids in the body and due to its endogenous de novo lipogenesis [5]. Still, based on long-term trial evidence, the palmitic acid content of CE also reflects dietary intake [4]. Odd-chain fatty acids (15:0 and 17:0) have been used as biomarkers for dairy fat intake, but they are also produced from gut-derived propionate [6] and exist in fish [7,8,9]. Moreover, the measured contents of given fatty acids in different biomarkers indicate only relative patterns of the fatty acids in the diet, and, e.g., incorporation of omega-3 polyunsaturated fatty acids (PUFAs) into various biomarkers may vary considerably [10,11]. Various fatty acid biomarkers reflect dietary intakes from a few days even to years, as does adipose tissue fatty acid composition [11]. Furthermore, well-known competition occurs between *n*-6 and *n*-3 PUFA; the higher the intakes and proportions of eicosapentaenoic acid (EPA) and docosahexaenoic acid (DHA) are, the lower the relative linoleic acid content of biomarkers [12,13]. Endogenous fatty acid metabolism is also quite complex, and some studies suggest, e.g., retroconversion of DHA to EPA and that DHA also interferes with linoleic acid metabolism [10]. Another example is that trans-fatty acid 18:1t incorporates markedly higher into TG and PL than CE and may result in a decreased conversion of linoleic acid to its more unsaturated metabolites [14]. Oleic acid (18:1*n*-9) is the dominant fatty acid in TGs, whereas linoleic acid (18:2*n*-6) is very abundant in CE fraction [15].

Postprandial fatty acid metabolism and fatty acid composition of biomarkers may be modified by other dietary components and factors related to glucose and insulin metabolism and cardio-metabolic health in general [16,17]. Finally, the fatty acid composition of various biomarkers is under genetic regulation [18,19]. Figure 1 summarizes the major factors affecting the fatty acid composition of serum lipids and cell membranes.

We became interested in the genetic regulation of fatty acid metabolism over 20 years ago when we examined whether polymorphism of the fatty acid binding protein 2 gene (*FABP2*) that is expressed in intestinal enterocytes modified postprandial lipemic response in humans [20]. In our oral fat-loading test, we examined postprandial triglyceride, chylomicron, and very-low-density lipoprotein TG in individuals homozygous for the Ala encoding allele (wild type) and individuals homozygous for the Thr54 allele variant. The Thr54 genetic variant was suggested to result in an increased absorption and processing of fatty acids in the intestine and showed an association with higher lipid oxidation and insulin resistance in Pima Indians [21] and with insulin resistance and increased intra-abdominal fat mass in Japanese men [22]. We found a markedly higher postprandial lipemic response in Finnish individuals homozygous for the Thr54 allele. Consequently, we also found an increased postprandial response of C14–C18 fatty acids in chylomicron and very-low-density lipoprotein (VLDL) TG in study persons homozygous for Thr54 variant [20]. However, we did not find any relative differences in the amounts of individual fatty acids introduced to these lipid fractions between these two genotype groups [23]. This particular genetic variant may also affect the postprandial TG content of high-density lipoprotein (HDL) in visceral obese individuals heterozygous for the Thr54 allele [24], and, interestingly, it may associate with reduced delta-6 desaturase (D6D) activity and lowered arachidonic acid (AA, 18:2*n*-6) content in obese children homozygous for Thr54 allele D [25]. In a clinical trial in patients with type 2 diabetes (T2D), individuals homozygous for the Thr54 allele showed an increased postprandial response of mono- and polyunsaturated fatty acids as an indication of increased fatty acid absorption [26]. In some other studies, postprandial lipemic response has not been related to this genetic polymorphism [16]. Nevertheless, these results on *FABP2* genetic variation, along with rapidly increasing knowledge about genes involved in fatty acid desaturation and elongation, indicate that fatty acid absorption and metabolism are under tight genetic regulation, and there may occur gene–diet interaction in fatty acid metabolism [19].

## 2. Methods

We carried out a literature survey of genes and diet in the context of the regulation of fatty acid compositions in PubMed. We used the following search term: (genotype OR gene OR fads OR elovl) AND fatty acid AND (diet OR dietary intake OR nutrition) AND (plasma OR erythrocyte membrane) AND human NOT animal. The search was performed on 3 July 2018 and it gave 553 hits altogether. Relevant articles were selected based on their abstracts (*n* = 32).

## 3. Genetic Variants of Fatty Acid Metabolism and Disease Risk

Figure 2 illustrates how a given genetic variant of fatty acid metabolism could modify endogenous fatty acid metabolism, their downstream metabolites, and, finally, risk of diseases. Principally, genetic variation may increase or decrease the activity of certain steps in endogenous fatty acid metabolism, affecting either desaturation or elongation processes. This effect is also modified by other factors, as shown in Figure 1. Altogether, modified fatty acid metabolism can be demonstrated by examining the fatty acid content of plasma and its components, cell membranes, or adipose tissue. In most studies [15,27,28], enzyme activities involved in fatty acid metabolism have been estimated by different ratios of certain fatty acids reflecting either desaturation or elongation of fatty acids to their longer chain metabolites (Figure 3 and Table 1). Furthermore, most of the known genetic variants (single-nucleotide polymorphisms, SNPs) associated with altered fatty acid metabolism are in fact genetic markers and their exact function is unknown. Even less is known about the interaction of genes and diet with regard to genetic regulation of endogenous fatty acid metabolism. In particular, genetic variants of the fatty acid desaturase (*FADS*) gene cluster in chromosome 11 (*FADS1*, *FADS2*) and elongases (e.g., *ELOVL2* and *ELOVL5*) are involved in the regulation of fatty acid metabolism.

Many studies suggest that delta-5-desaturase (D5D) activity (*FADS1*) is associated with a lower risk of T2D [29], while stearoyl-CoA desaturase 1 (SCD1) [27] and delta-6-desaturase (D6D, FADS2) variants are linked with insulin resistance and worsening of glucose tolerance or an increased risk of T2D [15,29,30,31,32]. In the longitudinal Finnish Diabetes Prevention Study, we confirmed the preventive effect of D5D activity on future T2D risk, and we showed that a higher insulin sensitivity may explain this finding [33]. Furthermore, based on the Metabolic Syndrome in Men (METSIM) study population where erythrocyte membrane fatty acids were analyzed, high estimated elongase activity was associated with a beneficial effect on glucose tolerance [30]. In another study carried out with the METSIM study population where the fatty acid compositions of serum lipids from TG, CE, and PL were analyzed, D6D activity predicted a worsening of glycemia, whereas elongase activity had an opposite effect on glycemia [15]. Furthermore, it is not surprising that genetic markers of the *FADS* cluster are also associated with lipid metabolism, in particular with HDL and triglyceride metabolism [34], but the data on cardiovascular disease (CVD) risk are lacking. In a meta-analysis studying associations between omega-3 fatty acid biomarkers and coronary heart disease (CHD), there was no significant interaction identified by *FADS* variant for incident CHD events [35]. Interestingly, in a genome-wide association study (GWAS) of Greenland Inuits, *ELOVL2* showed an association with sleep duration, age, and DNA methylation, and *ELOVL5* coding mutations may lead to spinocerebellar ataxia; as an example of epigenetic effect, epigenetic markers were associated with depression and suicide risk [19]. Disease per se may also modify enzyme activities involved in endogenous fatty acid metabolism. For instance, non-alcoholic steatohepatitis was found to affect desaturase enzyme activities in the liver in a cross-sectional study [36].

## 4. Genetic Regulation of Endogenous Fatty Acid Metabolism

Figure 3 summarizes the key enzymes regulating fatty acid metabolism in the body. Recently, many new genetic variants have been identified that may modify endogenous fatty acid metabolism and, consequently, the profile of fatty acid composition in different plasma components or tissues. In particular, the knowledge of genes participating in elongation processes has increased rapidly. We expect that new genetic variants linked with biomarkers reflecting the fatty acid composition of plasma components or cell membranes will be identified in the future. Some of these novel genetic variants are listed and briefly discussed in Table 2A–C and the next chapter.

There has been huge progress in research into fatty acid metabolism over the last 20 years. Both single-nucleotide polymorphism studies and GWAS have focused on enzymes regulating endogenous fatty acid metabolism (Figure 3). Specifically, these enzymes regulate both desaturation and elongation processes in fatty acid metabolism. Their genetic variations are associated with altered accumulation of long-chain PUFAs in the cell membranes of tissues and plasma lipid components [19]. Epigenetic variation of these genes may also play a role in this context [37,38]. In the present review based on a literature survey and our own studies, we focus on studies investigating the genetic regulation of plasma and membrane fatty acids in relation to the putative interaction between dietary fat and fatty acid incorporation into different biomarkers used to evaluate the quality of dietary fat in observational and intervention studies.

Table 2 summarizes the current knowledge about the genetic variants that have been associated with endogenous fatty acid metabolism. Most of the studies deal with individual SNPs of the *FADS* cluster (Table 2B) or elongases, and only some of them are based on GWAS (Table 2A). Among them, only a couple studies are true intervention trials (Table 2C), most without a specific hypothesis, and others are observational trials.

A GWAS based on the Invecchiare in Chianti (InCHIANTI) study population and Genetics of Lipid Lowering Drugs and Diet Network (GOLDN) study population used for replication observed that the *FADS* genetic cluster located in chromosome 11 was associated with AA, eisosadienoic acid (EDA), and EPA, and the *EVOLV2* genetic variant in chromosome 6 with EPA concentrations, but in the replication study, the association with EPA was not confirmed [18] (Table 2A). A few years earlier, two cross-sectional studies from Germany reported that haplotypes of the *FADS1* and *FADS2* region was associated with AA and many other longer-chain fatty acids of both the *n*-6 and *n*-3 series [39] (Table 2B); specifically, minor alleles of *FADS1*/*FADS2* showed mostly an association with decreased levels of plasma phospholipids. In a later study [40], associations between *FADS1*/*FADS2* haplotypes and fatty acids in phospholipids were replicated, and this study also showed similar associations with phospholipids of erythrocyte membranes, but only regarding omega-6 PUFAs (Table 2B). In 2011, Lemaitre and co-authors [41] published an important GWAS from five cohorts comprising altogether 8866 individuals of European ancestry, and smaller African, Chinese, and Hispanic populations were also examined. In line with previous studies, minor alleles of *FADS1* and *FADS2* were associated with higher ALA but lower EPA and docosapentaenoic acid (DPA), while minor alleles of *ELOVL2* were associated with higher EPA and DPA but lower DHA content. The results on *FADS1* were replicated in other ancestries examined. Furthermore, this study reported a novel association of DPA with several SNPs in *GCKR*.

With regard to interaction, ethnicity may also have an impact on gene–diet interactions. In a study on Inuits applying GWAS, several SNPs were examined in relation to erythrocyte membrane fatty acids [42]. Novel genes and polymorphisms that modified fatty acid composition were identified (Table 2A). This study also suggests that genetic and physiological adaptation to the intake of a diet rich in omega-3 PUFAs could happen with time.

## 5. Interaction between Genes and Dietary Fatty Acids

Most of the studies on interactions between genes and dietary fat in terms of the regulation of fatty acid composition of plasma or erythrocyte membranes are based on observational studies in various study populations. These studies have limitations relating to the accuracy of dietary data collection and possible weaknesses associated with controlling confounding factors (e.g., age, sex, ethnicity, lifestyle factors, liver status, and cardiometabolic health). Furthermore, genetic effect on fatty acid composition may vary depending on the fatty acid biomarker used. Therefore, intervention studies may give more comprehensive data on the variation of biomarker fatty acid composition according to the genetic background. Intervention studies also have limitations. They are laborious to carry out and without a pre-genotyped study population, the screening, especially regarding rare gene variants, is demanding. In our studies, we have been able to invite participants from the large METSIM cohort with wide genotype and phenotype data [43]. A big limitation related to experimental studies with a prior hypothesis is that in intervention studies, only one preselected genetic marker can be examined at a time. Furthermore, a highly experimental diet may not reflect the effects achieved with habitual diet, and adherence to an experimental diet may remain insufficient in free-living conditions in longer-term interventions. In large-scale observational studies, a multitude of genetic variations can be examined at the same time, and it is possible to combine the data collected from different study populations or the results may be confirmed in other study populations. 

Zietemann and co-authors examined the fatty acid composition of erythrocyte membranes and estimated desaturase activities in relation to the rs174546 *FADS* genotype variant in their cross-sectional study [44] (Table 2B). The estimated activities of *FADS1* and *FADS2* were strongly decreased in individuals with the minor allele, and in principle, fatty acid compositions reported were in line with earlier observations. Furthermore, this study described an interaction with diet, i.e., the dietary fatty acid *n*-6 to *n*-3 ratio was suggested to modify the association between the *FADS1* and *FADS2* genotype and estimated D5D activity calculated from measured fatty acids (see Table 1). In one study [45] (Table 2B), 309 pregnant women in the Netherlands were examined at the 36th gestational week and then one month postpartum. Both plasma phospholipids and milk fat composition postpartum were examined in relation to high fish or fish oil intakes. The results were divergent in phospholipids and milk: a higher omega-3 PUFA intake from fish or fish oil compensated for the lower DHA in plasma phospholipids irrespective of genotype, but the proportion of DHA in excreted milk remained unchanged in women who were homozygous for minor alleles of *FADS1*/*FADS2*. This study suggests that there may even be tissue-specific interactions regarding genes regulating fatty acid metabolism. Porenta and co-authors [46] (Table 2C) randomized 108 individuals with increased risk for colon cancer into the Mediterranean type diet or the Healthy Eating diet for 6 months. Serum and colonic mucosa fatty acid compositions were examined in relation to selected *FADS1*/*FADS2* alleles. In individuals with major alleles of the *FADS* cluster, interaction was suggested between the diets and colonic AA content that remained unchanged after the Mediterranean diet. In a small intervention study on putative interaction between fish oil supplementation and the *FADS* cluster, no significant interaction was found, but fish oil supplementation resulted in greater increases in erythrocyte EPA levels in minor allele carriers of *FADS1*/*FADS2* variants [47] (Table 2C).

In a large updated meta-analysis of the Cohorts for Heart and Aging Research in Genomic Epidemiology (CHARGE) consortium [48] (Table 2B), interaction between dietary PUFAs and 5 different genes affecting fatty acid composition was examined. However, no significant interactions were found after corrections. Interestingly, the results varied according to the compartments used. Specifically, regarding the *FADS1* interaction term for ALA, even opposite effects were found in proportions of fatty acids between plasma and erythrocyte membranes.

In a small randomized cross-over trial [49], individuals homozygous for the minor allele of *FADS1*/*FADS2* had a lower plasma AA and AA/LA ratio when compared with the major allele carriers after each diet, while *ELOVL2* had no effect on PUFAs. Furthermore, flaxseed oil, which is rich in ALA, resulted in increased plasma composition of EPA beyond that of major allele homozygotes consuming a typical “western” diet (Table 2C). While very sophisticated methodologies were applied, the study design was quite complicated in this particular study.

In one of our own cross-sectional studies [64], we reported a nominally significant gene–diet interaction between EPA in erythrocytes, CE and TG, and dietary intake of EPA in 962 men who were participating in the METSIM study (Table 2B). Interestingly, exclusion of fish oil supplement users strengthened the observed interaction with diet. We also confirmed that the minor allele of rs174550 of *FADS1* (C allele) was strongly associated with a lower hepatic mRNA expression, as observed recently by Wang et al. in their cross-sectional study [71]. Thus, we concluded that the observed interaction could be explained by divergent activity of the liver D5D enzyme in the genetic variants of *FADS* examined in our study.

Ideally, gene–diet interaction would be studied using an intervention design with participants with pre-selected genotypes. In our recent trial (Lankinen et al., Am J Clin Nutr 2018, in press), our aim was to test the hypothesis that the *FADS1* rs174550 genotype modifies the effect of dietary LA intake on the fatty acid composition of plasma lipids. Altogether, 59 men who were homozygotes for *FADS1* rs174550 SNP (TT or CC) completed the 4-week dietary intervention with a diet enriched in LA. During the 4-week intervention period, participants consumed their habitual diet with a supplement of 30 mL, 40 mL, or 50 mL (27–45 g) sunflower oil daily depending on their BMI. The doses of sunflower oil provided 17–28 g (6 E%) LA daily on top of the average intake of approximately 10–12 g (4.5 E%). The response in the proportion of AA in plasma phospholipids and cholesteryl esters differed between the genotype groups (Table 2C). The proportion of AA decreased in participants with the CC genotype, but remained unchanged (in PL) or decreased only slightly (in CE) in participants with the TT genotype. We also found that the *FADS1* genotype modified the lipid mediator profile (including eicosanoids and oxylipins) and inflammatory response, measured as serum high-sensitivity C-reactive protein, to an LA-rich diet.

## 6. Concluding Remarks

In this review, we aimed to summarize the current evidence regarding genes and dietary fatty acids in the regulation of the fatty acid composition of plasma lipids and erythrocyte membranes. The knowledge related to this topic has increased markedly during recent years, but there is no earlier review article compiling it together. The fatty acid composition of blood lipids and tissues is modified by dietary intake, but endogenous metabolism of fatty acids, which is strongly genetically regulated, also has an important role. In particular, genetic variants of the *FADS* gene cluster in chromosome 11 and elongases (*ELOVL*s) are involved in the regulation of fatty acid metabolism. In recent years, some new genetic variants have been shown to be associated with the fatty acid composition of plasma lipids or erythrocytes, and we expect that new variants will be identified in the near future. Many of the known genetic variants are quite common. Therefore, it is important to understand better how the enzymes regulating fatty acid metabolism, and the genes coding them, modify the effect of dietary intake of fatty acids on metabolism, low-grade inflammation, and metabolic diseases such as T2D. This understanding may help us to move towards personalized nutrition. It is also noteworthy that ethnicity may have an impact on gene–diet interactions. Most of the studies on interactions between genes and dietary fat in the regulation of fatty acid composition of plasma lipids are observational. Study designs and data collection should be carefully considered in the interpretation of the results. There are only a few intervention trials regarding this topic, and most of them were performed without pre-selected genotypes. Definite answers regarding true gene × diet interactions may need well-planned intervention studies with specific hypotheses and pre-selected genotypes.

## Figures and Tables

**Figure 1 nutrients-10-01785-f001:**
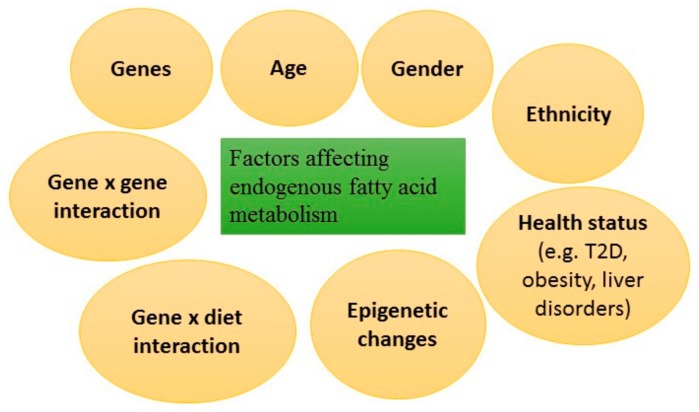
Factors affecting endogenous fatty acid metabolism. T2D, type 2 diabetes.

**Figure 2 nutrients-10-01785-f002:**
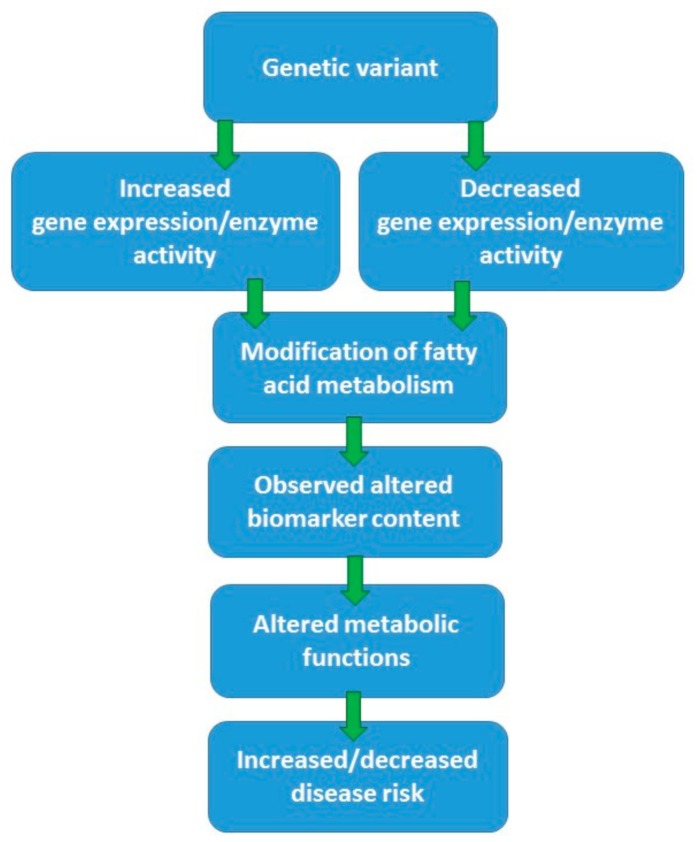
Impacts of genetic variants regulating fatty acid metabolism in the body.

**Figure 3 nutrients-10-01785-f003:**
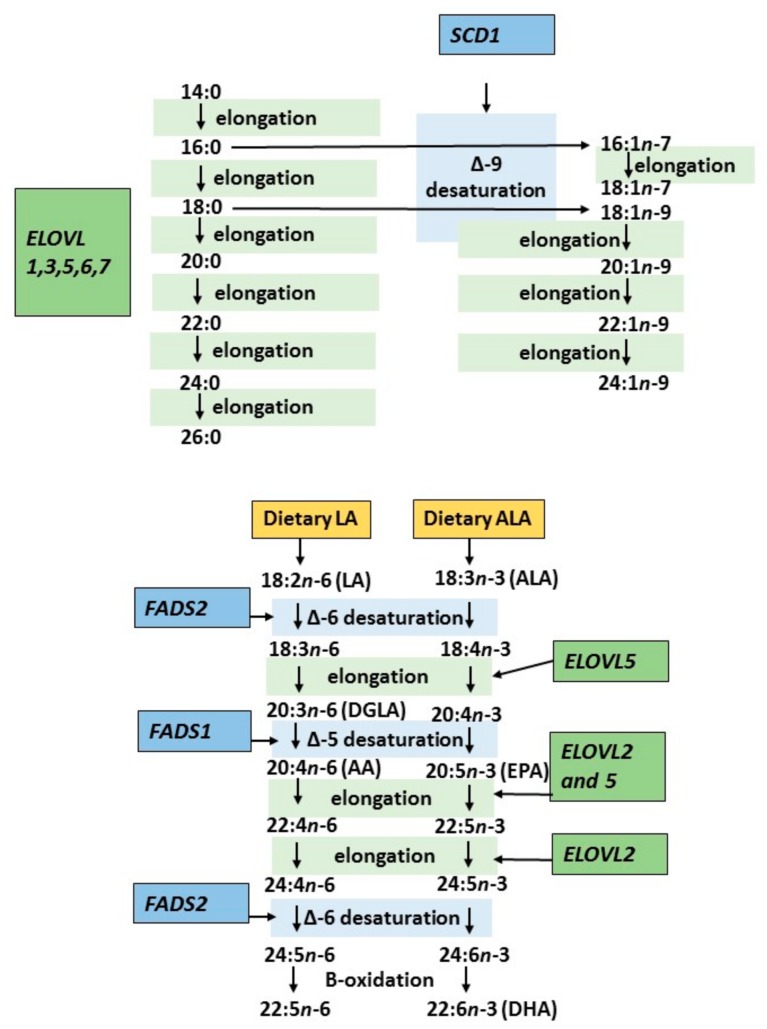
Simplified figure of the synthesis of fatty acids in the human body including key enzymes regulating fatty acid metabolism and genes coding them. AA, arachidonic acid; ALA, alpha-linolenic acid; DGLA, Di-homo-gamma linolenic acid; DHA docosahexaenoic acid; ELOVL, fatty acid elongase; EPA, eicosapentaenoic acid; FADS, fatty acid desaturase; LA, linoleic acid; SCD, stearoyl-CoA desaturase 1.

**Table 1 nutrients-10-01785-t001:** Fatty acid ratios used for the estimation of desaturase and elongase activities.

Estimated Desaturase/Elongase	Fatty Acid Ratio
Stearoyl-CoA desaturase 1 (SCD1)	16:1*n*-7/16:0
Delta-6-delta (D6D)	18:3*n*-6/18:2*n*-6
Delta-5-delta (D5D)	20:4*n*-6/20:3*n*-6
Elongase	18:1*n*-7/16:1*n*-7

**Table 2 nutrients-10-01785-t002:** Studies on the relation between genetic polymorphism and biomarkers of dietary fatty acids and reported interaction between diets and genetic variants regulating fatty acid metabolism. Part A includes genome-wide association studies (GWAS), Part B includes a priori selected genes, and Part C includes intervention studies.

**A) GWAS Studies**					
**Study**	**Study Population and Design**	**Fatty Acid Biomarkers Examined**		**Main Findings**	**Comment**
Tanaka T et al. PLoS Genet 2009. [18]	InCHIANTI Study (Chianti region of Tuscana, Italy, *n* = 1075) and GOLDN (predominantly Caucasian, *n* = 1076) replication study	PUFAs in plasma in CHIANTI study and in erythrocytes in GOLDN study		*FADS* genetic cluster marker (*FADS1*, *FADS2*, *FADS3*) in chromosome 11 associated with AA, EDA, and EPA, and *EVOLV2* genetic marker in chromosome 6 with EPA.	First GWAS with replication data. The *ELOVL2* SNP was associated with DPA and DHA but not with EPA in GOLDN replication study.
Lemaitre RN et al. PLoS Genet 2011. [41]	Five cohorts (*n* = 8866) of European ancestry (CHARGE consortium). In addition, African (*n* = 2547), Chinese (*n* = 633), and Hispanic ancestry (*n* = 661) study populations were examined.	Four major *n*-3 PUFAs (ALA, EPA, DPA, DHA) in plasma PL		Minor alleles of *FADS1* and *FAD*S2 associated with higher ALA, but lower EPA and DPA. Minor alleles of SNPs in *ELOVL2* were associated with higher EPA and DPA and lower DHA content.	A novel association of DPA with several SNPs in *GCKR* (glucokinase regulator) was reported. Results on *FADS1* were similar regardless of ancestry studied.
Wu et al. Circ Cardiovasc Genet 2013. CHARGE consortium [50]	European Ancestry (*n* = 8961)	Plasma levels of 16:0, 18:0, 16:1*n*-7, 18:1*n*-9		*ALG14* polymorphisms were associated with higher 16:0 and lower 18:0. *FADS1* and *FADS2* polymorphisms were associated with higher 16:1*n*-7 and 18:1*n*-9 and lower 18:0. *LPGAT1* polymorphisms were associated with lower 18:0. *GCKR* and *HIF1AN* polymorphisms were associated with higher 16:1*n*-7, whereas *PKD2L1* and a locus on chromosome 2 (not near known genes) were associated with lower 16:1*n*-7.	Polymorphisms in 7 novel loci were associated with circulating levels of ≥1 of 16:0, 18:0, 16:1*n*-7, 18:1*n*-9.
Guan et al. Circ Cardiovasc Genet 2014. [51]	White adults (*n* = 8631)	Total plasma or plasma PL *n*-6 PUFAs		Novel regions were identified on chromosome 10 associated with LA (rs10740118; near *NRBF2*); on chromosome 16 with LA, GLA, dihomo-GLA, and AA (rs16966952; *NTAN1*); and on chromosome 6 with adrenic acid after adjustment for AA (rs3134950; *AGPAT1*). Previous findings of the *FADS* cluster on chromosome 11 with LA and AA were confirmed.	
Dorajoo R et al. Genes Nutr. 2015. [52]	Singaporean Chinese population (*n* = 1361)	Plasma PUFAs		Genome-wide associations with ALA, all four *n*-6 PUFAs, and delta-6 desaturase activity at the *FADS1*/*FADS2* locus. These associations were independent of dietary intake of PUFAs.	Genetic loci that influence plasma concentrations of *n*-3 and *n*-6 PUFAs are shared across different ethnic groups.
Fumagalli M et al. Science 2015. [42]	Inuits (*n* = 191), European (*n* = 60), and Han Chinese (*n* = 44) individuals	Erythrocyte membrane fatty acids		*FADS1*, *FADS2*, *FADS3*, and SNPs 7115739, rs174570 (among others) had positive association with ETA, but negative associations with EPA and DPA; no effect on DHA content.	Novel genes and polymorphisms were identified in Inuits that may suggest genetic and physiological adaptation to a high-omega-3-PUFA diet; associations with height and weight were also found.
Lemaitre et al. J Lipid Res 2015. [53]	European ancestry (*n* = 10,129)	Plasma PL and Erythrocyte levels of VLSFA (20:0, 22:0, 24:0)		The *SPTLC3* (serine palmitoyl-transferase long-chain base subunit 3) variant at rs680379 was associated with higher 20:0. The *CERS4* (ceramide synthase 4) variant at rs2100944 was associated with higher levels of 20:0 and in analyses that adjusted for 20:0, with lower levels of 22:0 and 24:0.	*SPTLC3* is a gene involved in the rate-limiting step of de novo sphingolipid synthesis.
Mozaffarian et al. Am J Clin Nutr 2015. CHARGE consortium [54]	Meta-analysis of GWA studies (*n* = 8013)	Erythrocyte of PL trans fatty acids and 31 SNPs in or near the *FADS1* and *FADS2* cluster		Genetic regulation of cis/trans-18:2 by the *FADS1*/*2* cluster.	Trans fatty acids.
Tintle NL et al. Prostaglandins Leukot Essent Fatty Acids 2015. [55]	Framingham Offspring Study (*n* = 2633)	14 red blood cell fatty acids		Novel associations between (1) AA and *PCOLCE2* (regulates apoA-I maturation and modulates apoA-I levels), and (2) oleic and linoleic acid and *LPCAT3* (mediates the transfer of fatty acids between glycerolipids). Also, previously identified strong associations between SNPs in the *FADS* and *ELOVL* regions were replicated.	Multiple SNPs explained 8–14% of the variation in 3 high-abundance (>11%) fatty acids, but only 1–3% in 4 low-abundance (<3%) fatty acids, with the notable exception of DGLA acid with 53% of variance explained by SNPs.
de Oliveira Otto MC et al. CHARGE consortium. PLoS One 2018. [56]	Meta-analysis of GWA studies (*n* = 11,494); individuals of European descent	15:0, 17:0, 19:0, and 23:0 (OCSFA) in plasma PL and erythrocytes		SNP *MYO10* rs 13361131 associated with 17:0 level, *DLEU1* rs12874278 and rs 17363566 associated with 19:0 level. Using candidate gene approach, a few other SNPs also associated with 17:0 and 23:0 levels.	Circulating levels of OCSFA are predominantly influenced by nongenetic factors.
**B) Candidate Gene Studies**					
**Study**	**Study Population and Design**	**Fatty Acid Biomarkers Examined**	**Genes Examined**	**Main Findings**	**Comment**
Schaeffer L et al. Hum Mol Genet 2006. [39]	*N* = 727 from Erfurt, Germany, from The European Community Respiratory Health Survey I (ECRHS I), cross-sectional	PUFAs in plasma PL	Haplotypes of FADS1 and *FADS2* region	Haplotypes of *FADS1* and *FADS2* region associated with AA and many other long-chain *n*-6 and *n*-3 fatty acids (e.g., LA, GLA, and EPA and DPA.	Mostly decreased levels of PUFAs associated with minor alleles of *FADS1* and *FADS2*.
Xie L and Innis SM. J Nutr 2008. [57]	69 pregnant women in Canada and breast milk for a subset of 54 women exclusively breast-feeding at 1 month postpartum, cross-sectional	Plasma phospholipid and erythrocyte ethanolamine phosphoglyceride (EPG) (*n*-6) and (*n*-3) fatty acids	*FADS1*/*FADS2* rs174553, rs99780, rs174575, and rs174583	Minor allele homozygotes of rs174553 (GG), rs99780 (TT), and rs174583 (TT) had lower AA but higher LA in plasma phospholipids and erythrocyte EPG and decreased (*n*-6) and (*n*-3) fatty acid product/precursor ratios at 16 and 36 weeks of gestation.	Breast milk fatty acids were influenced by genotype, with significantly lower 14:0, AA, and EPA but higher 20:2(*n*-6) in the minor allele homozygotes of rs174553, rs99780, and rs174583 and lower AA, EPA, DPA, and DHA in the minor allele homozygotes of rs174575.
Rzehak P et al. Br J Nutr 2009. [40]	Bavarian Nutrition survey II, Germany, cross-sectional, (*n* = 163 and *n* = 535)	Phospholipid PUFA in plasma (*n* = 163), erythrocyte membranes (*n* = 535)	*FADS1* and *FADS2* haplotypes	Replication of *FADS1* and *FADS2* haplotypes associations in phospholipids (Schaeffer et al. 2006) and associations with PUFA in membranes.	Association with cell membranes was a novel finding. No associations with omega-3 PUFA.
Molto-Puigmarti C et al. Am J Clin Nutr 2010. [45]	KOALA Birth Cohort Study in the Netherlands. Plasma samples were collected at 36th gestational week in pregnant women (*N* = 309) and milk samples at 1 month postpartum.	Plasma phospholipids and milk DHA proportion	*FADS1* rs174561, *FADS2* rs174575, and intergenic rs3834458	A higher fish (or fish oil) intake compensated for the lower DHA proportions in plasma phospholipids irrespective of genotype but not in the milk from women with minor allele carriers of selected gene variants.	The study confirms earlier studies with regard to PUFA associations with minor allele carriers. Novelty of this study was gene–diet interaction regarding milk fat; DHA content remained unchanged with increasing fish/fish oil intake in women homozygous for minor allele.
Zietemann V et al. Br J Nutr 2010. [44]	A random sample of 2066 participants from the European Prospective Investigation into Cancer and Nutrition-Potsdam study, cross-sectional	Erythrocyte membrane fatty acids and estimated desaturase activity	rs174546 genetic variation (reflecting genetic variation in the *FADS1*/*FADS2* gene cluster)	Higher proportions of LA, EDA, and DGLA and lower proportions of GLA, AA, and DTA for the minor allele carriers. The estimated activities of *FADS1* and *FADS2* strongly decreased with minor T-allele.	Interaction with diet; dietary *n*-6/*n*-3 ratio was suggested to modify the association between the *FADS1*/*FADS2* genotype and the estimated D5D activity.
Dumont J et al. J Nutr 2011. [58]	European adolescents, HELENA study (*n* = 573), cross-sectional	Dietary intake of LA and ALA ALA, PUFA levels in serum PL Serum concentrations of TG, cholesterol, and lipoproteins	FADS1 rs174546	The associations between *FADS1* rs174546 and concentrations of PUFA, TG, cholesterol, and lipoproteins were not affected by dietary LA intake. Similarly, the association between the *FADS1* rs174546 polymorphism and serum phospholipid concentrations of ALA or EPA was not modified by dietary ALA intake. In contrast, the rs174546 minor allele was associated with lower total cholesterol concentrations and non-HDL cholesterol concentrations in the high-ALA-intake group but not in the low-ALA-intake group.	These results suggest that dietary ALA intake modulates the association between *FADS1* rs174546 and serum total and non-HDL cholesterol concentrations at a young age.
Merino et al. Mol Genet Metab. 2011. [59]	Toronto Nutrigenomics and Health study, (Caucasians *n* = 78, Asian, *n* = 69), cross-sectional	Plasma fatty acids	*FADS1* and *FADS2* genotypes (19 SNPs)	The most significant association was between the *FADS1* rs174547 and AA/LA in both Caucasians and Asians. Although the minor allele for this SNP differed between Caucasians (T) and Asians (C), carriers of the C allele had a lower desaturase activity than carriers of the T allele in both groups.	
Hong et al. Clin Interv Aging 2013. [60]	3 years follow-up study, nonobese men in South Korea (*n* = 122)	Serum PL PUFAs	near *FADS1* rs174537; *FEN1* rs174537G; *FADS2* rs174575 and rs2727270; *FADS3* rs1000778	The minor variants of rs174537 and rs2727270 were significantly associated with lower concentrations of long-chain PUFAs.	*FADS* polymorphisms can affect age-associated changes in serum phospholipid long-chain PUFAs, Δ5-desaturase activity, and oxidative stress.
Roke K et al. Prostaglandins Leukot Essent Fatty Acids 2013. [61]	Cross-sectional study, healthy young adults in Canada (*n* = 878)	Plasma levels of LA, GLA, DGLA, and AA	*FADS1*/*2* rs174579, rs174593, rs174626, rs526126, rs968567 and rs17831757	Several SNPs were associated with circulating levels of individual FAs and desaturase indices, with minor allele carriers having lower AA levels and reduced desaturase indices.	A single SNP in *FADS2* (rs526126) was weakly associated with hsCRP.
Huang et al. Nutrition 2014. [62]	T2DM patients (*n* = 758) and healthy individuals (*n* = 400) in Han Chinese, cross-sectional	Erythrocyte PL Fatty acids	Genetic variants in the *FADS* gene cluster	Minor allele homozygotes and heterozygotes of rs174575 and rs174537 had lower AA levels in healthy individuals. Minor allele homozygotes and heterozygotes of rs174455 in *FADS3* gene had lower levels of DPA, AA, and Δ5desaturase activity in patients with T2DM.	
Smith CE et al. Mol Nutr Food Res 2015. [48]	Updated meta-analysis of CHARGE consortium (*n* = 11,668) evaluating interactions between dietary PUFAs and selected genetic variants of 5 genes	Total plasma, phospholipids or erythrocyte membranes ALA, EPA, DHA, and DPA Dietary PUFA	*FADS1* rs174538 and rs174548; *AGPAT3* rs7435; *PDXDC1* rs4985167; *GCKR* rs780094; *ELOVL2* rs3734398	Primary aim was to examine gene–diet interactions regarding PUFAs. No significant interactions were found after corrections.	Fatty acid compartments affected the results, and, e.g., *FADS1* interaction terms for dietary ALA vs plasma phospholipids (negative) and erythrocytes (positive) were opposite.
Andersen et al. PLoS Genet 2016. [63]	Cross-sectional, Greenlanders (*n* = 2626)	22 FAs in the PL fraction in erythrocytemembranes	*ACSL6* rs76430747; *DTD1* rs6035106; *CPT1A* rs80356779; *FADS2* rs174570; *LPCAT3* rs2110073; *CERS4* rs11881630	Novel loci were identified on chromosomes 5 and 11, showing strongest association with oleic acid (*ACSL6*) and DHA (*DTD1*), respectively. For a missense variant (in *CPT1A*), a number of novel FA associations were identified; the strongest with 11-eicosenoic acid.	Novel loci associating with FAs in the PL fraction of erythrocytemembranes were identified in Greenlanders. For variants in *FADS2*, *LPCAT3*, and *CERS4*, known FA associations were replicated.
Takkunen M et al. Mol Nutr Food Res 2016. [64]	962 men from the METSIM study and Kuopio Obesity Surgery Study participants (*n* = 240) in Finland, cross-sectional	Fatty acid composition in erythrocyte and plasma PL, CE, and TG	Hepatic expression of *FADS1* (rs174547/rs174550)	A common *FADS1* variant (rs 174550) showed nominally significant gene–diet interactions between EPA in erythrocytes, and plasma CE and TG and dietary intakes.	Minor allele (C) of *FADS1* (rs174547) was strongly associated with reduced hepatic mRNA expression. High intake of EPA and DHA may reduce D5D activity in the liver.
de la Garza Puentes A et al. PLoS ONE 2017. [65]	PREOBE cohort in Spain (*n* = 180), 24 weeks of gestation, cross-sectional	Plasma PL FAs	7 SNPs in *FADS1*, 5 in *FADS2*, 3 in *ELOVL2* and 2 in *ELOVL5*	Normal-weight women who were minor allele carriers of *FADS* SNPs had lower levels of AA, lower AA/DGLA and AA/LA indices, and higher levels of DGLA compared to major homozygotes. Among minor allele carriers of *FADS2* and *ELOVL2* SNPs, overweight/obese women showed a higher DHA/EPA index than the normal-weight group.	Maternal weight modifies the effect of genotype on FA levels.
Guo H et al. Lipids Health Dis 2017. [66]	951 Chinese adults, cross-sectional	Plasma PL FAs *FADS1* rs174547	*FADS1* rs174547	The rs174547 C minor allele was associated with a higher proportion of LA, lower AA and DHA, as well as lower delta-6-desaturase and delta-5-desaturase activities.	Confirms earlier finding in Chinese population.
Kim et al. Prostaglandins Leukot Essent Fatty Acids 2018. [67]	Three-year prospective cohort study in Korea, 287 healthy subjects	Plasma PUFA levels	*FADS1* rs174547	The minor allele of the *FADS1* rs174547 associated with age-related decrease in the EPA/AA ratio among overweight subjects.	The minor allele of the *FADS* 1 rs174547 associated with increase in arterial stiffness among overweight subjects.
Li et al. Am J Clin Nutr 2018. [68]	1504 healthy Chinese adults, cross-sectional	Plasma PUFA concentration	FADS2 rs66698963	The rs66698963 genotype is associated with AA concentration and AA to EPA+DHA ratio.	Genotype also affected triglyceride and HDL cholesterol concentrations.
**C) Intervention Studies**					
**Study**	**Study Population and Design**	**Fatty Acid Biomarkers Examined**	**Genes Examined**	**Main Findings**	**COMMENT**
Al-Hilal M et al. J Lipid Res 2013. [69]	RCT in United Kingdom (*n* = 310) Supplementation of EPA + DHA 1) 0.45 g/day 2) 0.9 g/day 3) 1.8 g/day 4) placebo for 6 months	Plasma and erythrocyte PUFAs	*FADS1*/*FADS2* rs174537, rs174561, and rs3834458	s174537, rs174561, and rs3834458 in the *FADS1*–*FADS2* gene cluster were strongly associated with proportions of LC-PUFAs and desaturase activities estimated in plasma and Ery. In a randomized controlled dietary intervention, increasing EPA and docosahexaenoic acid DHA intake significantly increased D5D and decreased D6D activity after doses of 0.45, 0.9, and 1.8 g/day for six months. Interaction of rs174537 genotype with treatment was a determinant of D5D activity estimated in plasma.	Different sites at the *FADS1*–*FADS2* locus appear to influence D5D and D6D activity, and rs174537 genotype interacts with dietary EPA+DHA to modulate D5D.
Gillingham LG et al. Am J Clin Nutr 2013. [49]	36 hyperlipidemic individuals in Canada, randomized cross-over design with 3 experimental diets for 4 weeks 1) Flax seed oil 2) High canola oil 3) Western diet Only a few persons with minor allele of a given gene	Plasma FAs and (U-^13^C) ALA metabolism	SNPs for *FADS1*, *FADS2* and *ELOVL2*	Subjects homozygous for the minor allele of *FADS1*/*FADS2* had lower plasma composition of AA and AA/LA ratio in comparison with the major allele carriers after consumption of each experimental diet. *ELOVL2* had no effect on PUFAs.	Increasing ALA intake with a diet enriched in flaxseed oil in minor allele homozygotes resulted in an increased plasma composition of EPA beyond that of major allele homozygotes consuming a typical western diet.
Porenta SR et al. Cancer Prev Res (Phila) 2013. [46]	108 individuals with increased risk of colon cancer in USA, RCT for 6 months with two intervention diets: 1) Mediterranean type (MedD) 2) Heathy Eating Diet	Serum and colonic mucosa fatty acids	*FADS1*/*FADS2* minor allele SNPs (rs174556, rs174561, rs174537, rs3834458)	At 6 months, an increase in colonic AA in the Healthy Eating diet arm was found, while colon AA concentrations remained fairly constant in the MedD group in persons with major alleles in the *FADS1*/*2* gene cluster.	These results suggest gene–diet interaction in fatty acid metabolism related to different response to diets, but in individuals with major alleles of the *FADS* cluster.
Roke K and Mutch DM. Nutrients 2014. [47]	12 young men in Canada, 12 week intervention with fish oil capsules with 8 week wash-out; no control group	Fatty acid analysis from serum and erythrocytes	*FADS1*/*FADS2* (rs174537, rs174576)	Marked increase in serum and erythrocyte EPA and DHA. Elevation in RBC was sustained for 8 weeks during wash-out.	No significant gene × fish oil interaction, but % change in minor allele carriers of *FADS1*/*FADS2* had a greater increase in RBC EPA levels.
Scholtz SA et al. Prostaglandins Leukot Essent Fatty Acids 2015. [70]	Intervention, pregnant women (*n* = 205) in USA 1) Supplementation with 600 mg per day of DHA 2) Placebo for the last two trimesters of pregnancy	Plasma and RBC PL AA and DHA	*FADS1* rs174533 and *FADS2* rs174575	DHA but not the placebo decreased the AA status of minor allele homozygotes of both *FADS* SNPs but not major allele homozygotes at delivery.	
Lankinen et al. Am J Clin Nutr 2018. (in press)	Intervention, men with *FADS1* rs174550 TT or CC genotype (*n* = 59) in Finland High-LA diet for 4 weeks	Plasma PL and CE fatty acids	*FADS1* rs174550	There was a significant increase in the LA proportion in PL and CE in both genotype groups. A significant interaction between intervention and genotype was observed in AA, (decreased in CC genotype, but remained unchanged (in PL) or decreased only slightly (in CE) in TT genotype).	The response to higher LA intake in hsCRP was different between the genotypes. Individuals with the rs174550-TT genotype had a trend towards decreased hsCRP, while individuals with the rs174550-CC genotype had a trend towards increased hsCRP (significant diet × genotype interaction).

Abbreviations: AA, arachidonic acid (20:4*n*-6); ALA, alpha-linolenic acid (18:3*n*-3); DGLA; Di-homo-gamma linolenic acid (20:3*n*-6); DPA, docosapentaenoic acid (22:5*n*-3); EDA, eicosadienoic acid (20:2*n*-6); EPA, eicosapentaenoic acid (20:5*n*-3); Ery, erythrocyte membranes; ETA, eicosatetraenoic acid (20:4*n*-3); hsCRP; high-sensitivity C-reactive protein; GLA, gamma-linolenic acid (18:3*n*-6); LA, linoleic acid (18:2*n*-6), OCSFA; odd-chain saturated fatty acids; PL, phospholipid; PUFA, polyunsaturated fatty acid; RBC, red blood cell; RCT, randomized controlled trial; SNP, single-nucleotide polymorphism; T2D, type 2 diabetes; VLSFA, very-long-chain saturated fatty acids.

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
