# Peer review of "Genes and Dietary Fatty Acids in Regulation of Fatty Acid Composition of Plasma and Erythrocyte Membranes"

_nutrients, 2018, doi:10.3390/nu10111785_

Round 1
Reviewer 1 Report
Authors aim to determine the relation of diet and RBC and plasma fatty acid composition through literature survey, its an excellent compilation of literature review.
Line 60: We reported 20 years ago..
There should be a table for literature survey results and method used for screening e.g. Pubmed serach terms used (GWAS+FADS1/FADS2. RBC etc.) and number of results obtained
Although the tile claims this review correlates dietary fatty acid intake to fatty acid composition of plasma and RBCs the theme of the review is genomics and pathology centric by very nature of studying gene X diet interaction and should be reflected in modified title
Figure 2: Increased effect and decreased effect needs to be clarified, do authors mean activity of given gene/enzyme ?
Author Response
Authors aim to determine the relation of diet and RBC and plasma fatty acid composition through literature survey, its an excellent compilation of literature review.
Line 60: We reported 20 years ago.
Unfortunately, we do not understand what is suggested to be changed. Could you please clarify your comment?
There should be a table for literature survey results and method used for screening e.g. Pubmed serach terms used (GWAS+FADS1/FADS2. RBC etc.) and number of results obtained.
We thank the reviewer for this important comment. We have added a method section where we describe our PubMed search term and number of results obtained (Lines 88-94). In fact, two of the authors made independently a literature survey and checked the results together. If there are any relevant publications we have missed, we are ready to add them.
Although the tile claims this review correlates dietary fatty acid intake to fatty acid composition of plasma and RBCs the theme of the review is genomics and pathology centric by very nature of studying gene X diet interaction and should be reflected in modified title
We thank the reviewer for this comment. Only few studies investigating true gene x diet interactions have been published so far and therefore we had to broaden our focus to include also e.g. genomics. We have modified the title to be” Genes and dietary fatty acids in regulation of fatty acid composition of plasma and erythrocyte membranes”, which illustrates better the content of the review.
Figure 2: Increased effect and decreased effect needs to be clarified, do authors mean activity of given gene/enzyme?
This has been modified. As the reviewer suggested, we mean increased or decreased activity of gene or enzyme.
Reviewer 2 Report
The manuscript by Lanikinen and colleagues aims to summarize current evidence about the interaction between genes and diet in the regulation of fatty acids composition in plasma and erythrocyte membranes.
While the Authors defined their work as an extensive literature survey, I would define it as a review of current evidence, without a systematic approach. I really appreciated the topic of the manuscript since there is the need to summarize previous works, but I expected at least a description of methods used to conduct the review: the Authors did not explain keywords, inclusion and exclusion criteria, data extraction protocols etc.
Moreover, the Authors mainly focused on genetic variations in FADS genes, but mentions to other genes, as done for ELOVL and SCD1, might improve the quality of their work.
I also suggest to reorganize the manuscript using different paragraphs for the results on plasma and erythrocyte membranes.
In general, an extensive editing of language and style is recommended since I found some errors/typos (e.g. lines 32-33; 45-46; 83; 215-216) and some confusing paragraphs (215-219) .
In line 87, the Authors stated that Figure 2 illustrates how genetic variants could modify fatty acid metabolism. However, I did not find this Figure informative for the scope of the Authors.
In table 3 and throughout the text, the Authors are suggested to indicate the study design for each study. Moreover, I suggest to use two separate columns for fatty acids biomarkers and genes, indicating both genes and genetic variants for each study. Since the Authors reported that ethnicity may have an impact on gene-diet interactions, I also suggest to add a column indicating the ethnicity of study population.
In the concluding remarks sections, I think the manuscript could benefit from some discussion on the novelty and originality of the work.
Minor changes:
The Authors are suggested to check abbreviations (e.g. triglycerides) and to use round brackets for “e.g.” or “i.e”.
In line 114, CVD stand for cardiovascular disease.
Author Response
The manuscript by Lanikinen and colleagues aims to summarize current evidence about the interaction between genes and diet in the regulation of fatty acids composition in plasma and erythrocyte membranes.
While the Authors defined their work as an extensive literature survey, I would define it as a review of current evidence, without a systematic approach. I really appreciated the topic of the manuscript since there is the need to summarize previous works, but I expected at least a description of methods used to conduct the review: the Authors did not explain keywords, inclusion and exclusion criteria, data extraction protocols etc.
We thank the reviewer of this comment. It is true that this is a review of current evidence without systematic approach. We have now added our PubMed search terms and number of results obtained (lines 88-94). In fact, two of the authors made independently a literature survey and checked the results together. If there are any relevant publications we have missed, we are ready to add them.
Moreover, the Authors mainly focused on genetic variations in FADS genes, but mentions to other genes, as done for ELOVL and SCD1, might improve the quality of their work.
FADS genes have been widely studied and they have been associated with fatty acid proportion also in GWAS studies. ELOVL and SCD genes have also been studied quite extensively, but not that much. There are also some studies showing e.g. association between fatty acid proportions and ApoE4 or PPAR-alpha polymorphisms. Since these associations have not been shown in GWAS studies we decided to leave those studies out.
I also suggest to reorganize the manuscript using different paragraphs for the results on plasma and erythrocyte membranes.
Since fatty acid proportions of plasma lipids and erythrocyte membranes highly correlate with each other, we would like to discuss them together. We feel that by separating them into different paragraphs may lead to unnecessary repetition of the results.
In general, an extensive editing of language and style is recommended since I found some errors/typos (e.g. lines 32-33; 45-46; 83; 215-216) and some confusing paragraphs (215-219).
Thank you for these remarks. We have tried to clarify confusing paragraphs and correct errors.
In line 87, the Authors stated that Figure 2 illustrates how genetic variants could modify fatty acid metabolism. However, I did not find this Figure informative for the scope of the Authors.
We apologize our vague phrasing. We have clarified that this figure goes beyond fatty acid metabolism and illustrates how these genetic variants could modify endogenous fatty acid metabolism, their downstream metabolites and finally also risk of diseases. We think that this kind of the figure help readers who are not familiar with the issue to get general idea of the importance of the review.
In table 3 and throughout the text, the Authors are suggested to indicate the study design for each study. Moreover, I suggest to use two separate columns for fatty acids biomarkers and genes, indicating both genes and genetic variants for each study. Since the Authors reported that ethnicity may have an impact on gene-diet interactions, I also suggest to add a column indicating the ethnicity of study population.
We have now indicated the study design for each study throughout the text and Table 2. We have also separated fatty acid biomarkers and genes for two columns in Table 2 and indicated the ethnicity or the country where the research has been performed whenever it has been defined in the original article.
In the concluding remarks sections, I think the manuscript could benefit from some discussion on the novelty and originality of the work.
We have added few sentences related to the novelty and originality of the work.
Minor changes:
The Authors are suggested to check abbreviations (e.g. triglycerides) and to use round brackets for “e.g.” or “i.e”.
Abbreviations have been corrected.
In line 114, CVD stand for cardiovascular disease.
The word “disease” has been added.
Reviewer 3 Report
I am very confused about the structure of the whole review and what type of conclusion does the author tends to express. If the content of the review is to focus on the gene-diet interaction as the current title goes, then please remove any studies with only the main effects of the gene or fatty acids. If the authors intended to mention 3 things: 1) main effect of gene on fatty acids composition; 2) main effect of dietary fatty acids on the fatty acid composition; 3) the interaction between gene and dietary fatty acids on the fatty acid composition, then the most important thing I will suggest is to change the title to "Genes and dietary fatty acids in regulation of fatty acid composition of plasma and erythrocyte membranes"
Please make the sub-title of each part concise and clear, not like the sentence in the part 4.
The first 2 figures and Table 1 are useless and without informative and official editing. I will suggest to remove them.
Please correct the expression of gene name in the figure. You do not need to mention the "gene" after the gene name, which is enough to use to the italic style. Also, please use the full word of "chromosome" rather than "Chr". The GWAS should be "genome-wide association studies", not "gene wide association...". The text does have a lot technical and professional expressions need to be fixed.
Author Response
I am very confused about the structure of the whole review and what type of conclusion does the author tends to express. If the content of the review is to focus on the gene-diet interaction as the current title goes, then please remove any studies with only the main effects of the gene or fatty acids. If the authors intended to mention 3 things: 1) main effect of gene on fatty acids composition; 2) main effect of dietary fatty acids on the fatty acid composition; 3) the interaction between gene and dietary fatty acids on the fatty acid composition, then the most important thing I will suggest is to change the title to "Genes and dietary fatty acids in regulation of fatty acid composition of plasma and erythrocyte membranes"
We thank the reviewer for this excellent suggestion. We have now changed the title as suggested. Quite few studies investigating true gene x diet interactions have been published so far and therefore we had to broaden our focus to include all the parts mentioned above.
Please make the sub-title of each part concise and clear, not like the sentence in the part 4.
Sub-titles have been clarified and shortened. We have also modified the order of the text to follow better the sub-titles.
The first 2 figures and Table 1 are useless and without informative and official editing. I will suggest to remove them.
We think that for many readers they might help understand the whole picture of the topic. However, if the editor strongly suggests removing them, we consider to do that.
Please correct the expression of gene name in the figure. You do not need to mention the "gene" after the gene name, which is enough to use to the italic style. Also, please use the full word of "chromosome" rather than "Chr". The GWAS should be "genome-wide association studies", not "gene wide association...". The text does have a lot technical and professional expressions need to be fixed.
We thank the reviewer for these important remarks. We have corrected these.
Round 2
Reviewer 2 Report
The Authors have accomplished all my suggestions improving their manuscript
Reviewer 3 Report
All the comments have been revised.